# Killing of a Multispecies Biofilm Using Gram-Negative and Gram-Positive Targeted Antibiotic Released from High Purity Calcium Sulfate Beads

**DOI:** 10.3390/microorganisms11092296

**Published:** 2023-09-12

**Authors:** Kelly Moore, Anthony Li, Niraj Gupta, Tripti Thapa Gupta, Craig Delury, Sean S. Aiken, Phillip A. Laycock, Paul Stoodley

**Affiliations:** 1Department of Microbial Infection and Immunity, The Ohio State University, Columbus, OH 43210, USA; kelly.e.moore20@gmail.com (K.M.); li.5906@osu.edu (A.L.); gupta.1107@osu.edu (N.G.); thapatripti.bme@gmail.com (T.T.G.); 2Biocomposites Ltd., Keele Science Park, Keele, Staffordshire ST5 5NL, UK; craig_delury@hotmail.com (C.D.); sa@biocomposites.com (S.S.A.); pl@biocomposites.com (P.A.L.); 3Department of Orthopedics, The Ohio State University, Columbus, OH 43203, USA; 4Department of Microbiology, The Ohio State University, Columbus, OH 43210, USA; 5National Centre for Advanced Tribology at Southampton (nCATS), Department of Mechanical Engineering, University of Southampton, Southampton SO17 1BJ, UK; 6National Biofilm Innovation Centre (NBIC), University of Southampton, Southampton SO17 1BJ, UK

**Keywords:** biofilm, calcium sulfate, orthopedic infection, multispecies infection

## Abstract

Background: Multispecies biofilm orthopedic infections are more challenging to treat than mono-species infections. In this in-vitro study, we aimed to determine if a multispecies biofilm, consisting of Gram positive and negative species with different antibiotic susceptibilities could be treated more effectively using high purity antibiotic-loaded calcium sulfate beads (HP-ALCSB) containing vancomycin (VAN) and tobramycin (TOB) in combination than alone. Methods: Three sets of species pairs from bioluminescent strains of *Pseudomonas aeruginosa* (PA) and *Staphylococcus aureus* (SA) and clinical isolates, *Enterococcus faecalis* (EF) and *Enterobacter cloacae* were screened for compatibility. PA + EF developed intermixed biofilms with similar cell concentrations and so were grown on 316L stainless steel coupons for 72 h or as 24 h agar lawn biofilms and then treated with HP-ALCSBs with single or combination antibiotics and assessed by viable count or bioluminescence and light imaging to distinguish each species. Replica plating was used to assess viability. Results: The VAN + TOB bead significantly reduced the PA + EF biofilm CFU and reduced the concentration of surviving antibiotic tolerant variants by 50% compared to single antibiotics. Conclusions: The combination of Gram-negative and positive targeted antibiotics released from HP-ALCSBs may be more effective in treating multispecies biofilms than monotherapy alone.

## 1. Introduction

Periprosthetic joint infections (PJI) are a major complication following total knee or hip arthroplasty (TKA and THA) affecting between 1 and 2% of patients. While one Canadian study found a slight decrease in PJI over 15 years the high projected growth of TKA and THAs is estimated to result in nearly 100,000 PJI’s by 2030, costing over $1.1 billion [1,2]. PJIs have high societal and personal cost to the patient, the family and surgeon since they often require multiple surgical procedures and amputation is not uncommon. One reason PJIs are a challenge to treat is due to the formation of biofilm by the infecting pathogens [3]. These communities of bacteria form an extracellular matrix that make them highly tolerant of antimicrobial agents and protects them from physical disruption and host innate immunity. Multispecies biofilms are reported in approximately 15% of PJI cases and are more difficult to identify due to a number of factors including the inability of some species to grow independently in pure culture, variable growth times and nutritional needs [4,5,6,7,8,9]. Multispecies biofilms are also more difficult to treat as they can contain both Gram-positive and Gram-negative organisms with varying antibiotic susceptibilities [10,11].

Although multispecies biofilms are present in many infections, in-vitro biofilm models tend to consider a single species challenged with a single antibiotic [12,13,14,15,16]. However, species and antibiotic interactions can influence susceptibilities. Tavernier et al. found certain antibiotics increased the protection of *S. anginosus*, because of the stress response of *S. aureus* secreting a protective agent into the medium [16], showing that the susceptibility of a bacterial pathogen grown in pure culture may not reflect that when present in a multispecies biofilm. A combination of vancomycin and gentamicin or tobramycin are commonly incorporated into polymethylmethacrylate (PMMA) bone cement by surgeons to provide broad spectrum coverage, including of methicillin resistant *S. aureus* (MRSA) in PI cases. However, PMMA is non absorbable and after the initial burst of release much of the antibiotic remains trapped in the PMMA, and higher concentrations compromise mechanical stability and can even act antagonistically in terms of elution characteristics [17]. Absorbable mineral bone void fillers do not have this diffusion limitation since they completely dissolve over time releasing all of the antibiotics. Generally, in vitro studies of the antimicrobial activity of antibiotic loaded bone cements and absorbable bone void fillers against multiple species of pathogens, as biofilms or in flask cultures, do so against each species individually [18]. Previously, we have shown single species biofilms of both Gram-positive and Gram-negative species could be reduced or eradicated using dual-antibiotics released from high purity antibiotic loaded calcium sulfate bead (HP-ALCSB) bone void fillers in-vitro [19]. However, there are very few studies assessing the activity of dual antibiotic loaded cement and absorbable mineral fillers against multispecies biofilm. Therefore, we aimed to determine the efficacy of dual-antibiotics to target both Gram-positive and Gram-negative pathogens when released from HP-ALCSB in-vitro compared to individual antibiotics alone. First, we screened for compatible partner pairs from strains of *Pseudomonas aeruginosa*, *Staphylococcus aureus*, *Enterococcus faecalis* and *Enterobacter cloacae*. Our criteria for selection of the dual species were: (1) to include a Gram negative and Gram positive species to provide different antibiotic susceptibilities, (2) a rod and a cocci so that we could differentiate them by scanning electron microscopy (SEM), and (3) each species to establish in the biofilm in approximately equal numbers. Based on these criteria we selected *P. aeruginosa* and *E. faecalis*. Biofilms were grown on 316L stainless steel coupons or agar plates and challenged with HP-ALCSBs loaded with vancomycin alone, tobramycin alone or a combination of vancomycin and tobramycin. Antibiotic efficacy was assessed by viable plate count for the coupons and visual inspection of the zones of clearing on the agar plates.

## 2. Materials and Methods

### 2.1. Bacteria

Bioluminescent strains of *Pseudomonas aeruginosa* PAO1 (PA; XEN41) and *Staphylococcus aureus* USA300 (SA; SAP231) and clinical isolates from a biofilm infected abdominal repair mesh, *Enterococcus faecalis* BS374 (EF) and *Enterobacter cloacae* BS420 (EC) were used [20]. We chose one partner to be bioluminescent to help to distinguish strains on a single agar plate. For microscopy the species could be identified on the basis of morphology (Gram-positive cocci and Gram-negative rods). EC-BS420 and EF-BS374 were isolated from a multispecies biofilm infection associated with surgical repair mesh [20,21]. For each experiment, an isolated colony of each species was used to inoculate brain heart infusion media (BHI; Becton, Dickinson and Company, Sparks, MD, USA) and cultured overnight at 37 °C with shaking (Heracell 150i, 200RPM, Thermo Fisher Scientific, Waltham, MA, USA). 

### 2.2. Biofilm Development and Preliminary Screening for Compatible Gram Positive and Negative Pairs

Initially, we sought to determine which organisms would grow together in relatively equal cell densities in the biofilm. An overnight culture of each species was diluted to a concentration of ~1 × 10^5^ cells based on optical density in BHI broth. 5 mL was added to a 12-well microtiter plate containing a Stainless Steel 316L (316L-SS) coupon (BioSurface Technologies, Bozeman, MT, USA) because of its common use in orthopedics [22]. For multispecies cultures, we diluted overnight cultures of two individual species into BHI broth to a total concentration of ~1 × 10^5^ cells. Combinations were SA-SAP231 and PA-XEN41, SA-SAP231 and EC-BS420, and EF-BS374 and PA-XEN41. 5 mL of the multispecies culture was added to a 316L-SS coupon in a 12-well microtiter plate to achieve a final initial concentration of each species of ~2.5 × 10^5^ CFU. For both monospecies and multispecies biofilms, the 12-well microtiter plates were incubated dynamically in a 5% CO_2_ 37 °C shaker (200RPM,) for 72 h with media exchanges every 24 h. After 72 h, each coupon was gently rinsed three times in 5 mL of phosphate-buffered saline (PBS) to remove planktonic cells then transferred to a 50 mL conical tube containing 10 mL PBS. To suspend bacterial cells into solution for plating, we used an adaptation of a standard sonication method [23]. Each conical tube was sonicated for 5 min in a water bath sonicator (Fisher Scientific FS7652H), then vortexed for 30 s. This was repeated three times. The final solution was serially 1-fold diluted in PBS and plated on BHI agar for CFU counts. The detection limit was defined as one countable colony. The relative proportions of each species in the biofilm were found by counting colonies, which were identified based on size, shape, color, and bioluminescence (IVIS 100, Xenogen, Waltham, MA, USA). After our initial screen we chose the EF and PA partner pair for the remainder of the study.

### 2.3. Scanning Electron Microscopy (SEM)

Scanning electron microscopy was used to confirm the presence of both species in the EF + PA biofilm on the 316L-SS coupons which were distinguished by their cocci and rod morphologies. The coupons were soaked in a prefixing agent of 2.5% glutaraldehyde in 0.2 M cacodylate buffer (pH 7.4) for 24 h at room temperature before then rinsed with 0.2 M cacodylate buffer three times. After the final rinse, the coupons were dehydrated by placing in increasing concentrations of ethanol series (70%, 90%, and 100%) three times each for 20 min followed by dehydration with hexamethyldisilazane (HMDS). The coupons were then gold-palladium sputter coated before SEM imaging (Quanta 200, FEI, Hillsboro, OR, USA). Bacteria in the SEM micrographs were identified by eye based on size (rods and cocci) and shape (~1 µm dia. and (~1 µm width and 2–3 µm length for the rods) and false colored using photo-editing software (Adobe Photoshop version 22.4.2) to afford better contrast with the background.

### 2.4. Minimum Inhibition Concentration (MIC) Using E-Test Strips

To determine the susceptibility of EF and PA against vancomycin and tobramycin, antibiotics commonly used in bone cement and mineral bone void fillers to treat PJIs [24] we used MIC Test Strips (MTS Synergy Applicator System, Liofilchem, Waltham, MA, USA). Overnight cultures of individual species were smeared across BHI agar (1.5%) plates using a sterile loop to create a confluent lawn. E-test strips were placed in the center of individual agar plates and lightly pressed down to confirm the antibiotic gradient was in contact with the plate and incubated at 37 °C, 5% CO_2_ for 24 h. Our PA strain was resistant to VAN, but susceptible to TOB at 1 ug/mL. Our EF strain was susceptible to VAN at 1.5 ug/mL, but resistant to TOB at 12 ug/mL. 

### 2.5. Preparation of High Purity Antibiotic Loaded Calcium Sulfate Beads

HP-ALCSB (4.8 mm diameter) were prepared using Stimulan^®^ Rapid Cure (SRC) (Biocomposites Ltd., Staffordshire, UK) according to manufacturer instructions [25]. 20 g of CaSO_4_ powder was combined with either 1000 mg VAN, 240 mg TOB, or a combination of both (1000 mg VAN + 240 mg TOB) (Gold Biotechnology Inc., Olivette, MO, USA). Unloaded beads were used as negative controls. For the latter, the dry components were fully mixed, then 6 mL of the liquid mixing solution was added. All components were stirred for 30 s until the mixture became a paste, which was pressed into a mold mat (Biocomposites Ltd., Staffordshire, UK). The paste was set for 10–15 min at 20 °C before being removed as solid beads.

### 2.6. Antibiotic Challenge

EF and PA biofilms were grown on 316L-SS coupons for 72 h and gently rinsed as described above to remove planktonic bacteria. After rinsing, each coupon was placed in 5 mL of fresh BHI media broth with one HP-ALCSB loaded with either VAN alone, TOB alone, VAN + TOB or an unloaded bead. The biofilms were incubated on an orbital shaker at 200 RPB for 72 h at 37 °C, 5% CO_2_ as previously described [19]. After 72 h, each coupon was transferred to a 50 mL conical tube containing 10 mL PBS. Coupons were sonicated for 5 min, then vortexed for 30 s three times. CFUs were performed on the sonicate using the drop plate method.

### 2.7. Agar Plate Model

A previously established method to assess the potency of antibiotic released from HP-ALCSB to kill lawn biofilms by agar diffusion was used to simulate biofilms grown on soft tissue [26,27,28]. Equal concentrations of overnight cultures of EF-BS374 and PA-XEN41 were diluted and added to 15 mL BHI to a final concentration of ~10^5^ cells. 100 µL was plated onto BHI agar then incubated at 37 °C for 24 h before placing a HP-ALCSB loaded with VAN, TOB, VAN + TOB or an unloaded bead negative control at the center. Plates were imaged via an in-vitro imaging system (IVIS 100, Xenogen, Waltham, MA, USA) and light photos (iPhone camera) every 24 h for 9 d. On the 9th day, plates were replica plated onto fresh BHI agar plates without antibiotics to determine the viability of each species within the zones of biofilm clearing. Replica plating was performed using a sterile, cotton velveteen square (150 × 150 mm) which was aseptically draped over a PVC replica plater and locked in place with an aluminium ring. The square was gently stamped from the original plate onto a fresh sterile BHI agar plate to transfer colonies. Replica plates were imaged via IVIS and light photos after 24 h of incubation at 37 °C. This technique allowed us to determine the zones on the plate where the biofilm lawn had been completely eradicated (the zone of killing surrounding the bead), the zone of the confluent growth (zone of biofilm) beyond the zone of killing and the intermediate zone where most bacteria had been killed (>99.999%), but persister cells and other antibiotic tolerant variants survived. Using NIH Image FIJI, the diameters of the three zones were measured in millimetres on the replica plate image using the diameter of the plate as an internal scale. Additionally, in the intermediate zone the number of surviving antibiotic tolerant variant colonies were counted. 

### 2.8. Statistics

For a more accurate quantification of central tendency CFU date were first log_10_ transformed and presented graphically as log_10_ of the geometric mean ± SD. One-way ANOVA was conducted to compare the antibiotic treatments amongst the two species using GraphPad Prism, version 9 for MAC (GraphPad Software, San Diego, CA, USA). Differences were considered statistically significant for *p* < 0.05.

## 3. Results

### 3.1. Preliminary Screen for a Compatible Pair Multispecies Biofilm

We chose the EF and PA partner for the study because they populated the biofilm in relatively equal concentrations (Log_10_ 9.44 CFUs and Log_10_ 9.23 CFUs respectfully) compared to SA + PA and SA + EC (Figure 1, Figure 2 and Figure 3). Additionally, the two species in this combination could be easily identified on the same plate (Figure 4) and are more commonly found together clinically [29,30]. SA + EC and SA + PA both formed multispecies biofilms with our culture method, however, there was greater variation in CFUs between the species (Figure 2 and Figure 3).

The SEM images confirmed that EF and PA cells physically occupied the same locations in the biofilm demonstrating an integrated multispecies biofilm (Figure 5).

### 3.2. Antibiotic Challenge of EF + PA Biofilms Grown on 316L-SS from HP-ALCSB as Mono and Dual Treatments

Three-day dual-species biofilms were grown on 316L-SS, washed, then treated with a single HP-ALCSB or unloaded bead for three days. There were significant differences between VAN, TOB and VAN + TOB antibiotic treatments (Figure 6). VAN alone reduced CFUs by approximately 3 logs for EF-BS374 and PA-XEN41 individually and the total biofilm by 2.2 logs compared to controls. Similarly, TOB alone reduced CFUs of EF-BS374 approximately 3 logs and the total biofilm 2.7 logs compared to controls. PA-XEN41 was cleared from the multispecies biofilm. VAN + TOB together reduced CFUs of both PA-XEN41 and EF-BS374 by approximately 8.5 and 7.7 logs respectively and reduced the total biofilm by 6.6 logs; a statistically significant larger reduction than using either antibiotic alone for both species and TOB alone for EF-BS374.

### 3.3. Antibiotic Challenge on an EF + PA Biofilm Grown on Agar from HP-ALCSB as Mono and Dual Therapy

The agar plate model provided a means to macroscopically visualize the impact various antibiotics had on multispecies biofilms. PA-XEN41 colonies were larger, creamy yellow and irregularly shaped with lobate margins whereas EF-BS374 colonies were smaller, white, with undulate margins (Figure 7), as well as being bioluminescent (Appendix A). Before adding HP-ALCSB it was confirmed each plate displayed intermixed growth of both species. Colonies of PA-XEN41 treated with TOB alone appeared in the intermediate zone at the edge of the cleared lawn area after day 4 and were more distinct on day 5 (Figure 7). Previously we identified these as antibiotic tolerant colonies [31]. Replica plating on day 9, showed that VAN + TOB treatment of EF-BS374 and PA-XEN41 monospecies biofilms cleared the lawn biofilm, leaving a small subset of individual colonies behind. VAN treatment alone completely cleared EF in the multispecies biofilm, however a lawn of PA remained. The zone of killing achieved by VAN + TOB was larger than by TOB alone, but not significantly different (Figure 8). However, when using VAN + TOB, the number of antibiotic tolerant variants were reduced by almost 50% compared to TOB alone (60 to 113 respectively) (Table 1, Figure 9). Antibiotic tolerant variant colonies in EF or EF + PA biofilms were not observed with VAN alone.

## 4. Discussion

PJIs are a serious complication after total knee or hip arthroplasty due to difficulty eradicating biofilm even after irrigation, debridement, and even total revision since they require extended antibiotic treatment at high concentrations. Multispecies infections create an even more complex infection to treat as they more commonly include high virulent and multi-drug resistant organisms with lower success rates, more surgeries and longer periods of hospitalizations than monospecies infections [32,33]. In our study, we developed in-vitro multispecies EF and PA biofilm models with different antibiotic susceptibilities (EF was sensitive to VAN but resistant to TOB and vice versa for PA) on 316L-SS coupons or BHI agar plates to assess the efficacy of dual-antibiotic treatment released from HP-ALCSB (Table 1, Figure 2). SEM suggested that both species were intermixed in the biofilm SEM images suggested that the species were interspersed together within the biofilm rather than forming separate aggregates of each species., and each were actively growing on the 316LSS coupons as evidenced by dividing cells (Figure 5). Although the numbers of EF and PA in the biofilm were on the same order of magnitude (9.23 and 9.44 log10 CFU/cm^2^ respectively, PA was in greater abundance than EF, 2.7 × 10^9^ compared to 1.7 × 10^9^. This difference may be explained by different fitness levels in competition for space and nutrients.

When treated using either antibiotic from HP-ALCSB alone, the EF + PA multispecies biofilm on 316L-SS coupons was significantly reduced compared to the control (Figure 6). It is important to note we found a reduction in PA by VAN alone even though it is clinically resistant to VAN. This may be due to the very high concentrations achieved by local release from HP-ALCSB which is outside of conventional MIC testing or systemic achievement. A recent finding has also reported that under nutrient depletion conditions, which might be expected in the biofilm, *P. aeruginosa* becomes more sensitive to vancomycin down to or less than 64 µg/mL [34]. Also, interactions between species within a multispecies biofilm can result in a change of susceptibility to antibiotic treatment or biofilm structure [35,36,37,38]. For example, PA has been shown to have an increased susceptibility to other antibiotics in the presence of VAN [39]. In contrast, on agar plates, only PA remained after replica plating suggesting EF was either outcompeted or a combination of competition and antibiotic treatment allowed an overgrowth of PA. TOB alone produced similar results as VAN alone within the EF dominated portion of the multispecies biofilm; however, it reduced the concentration of the PA below detectable limits (Figure 6). Even so, using TOB alone in the agar plate model left small populations of PA antibiotic tolerant variants remaining in the initially cleared biofilm lawn zone (Figure 7) as seen previously [31]. This suggests that although TOB alone was able to reduce similar concentrations on coupons and showed more clearance of biofilm on agar than the antibiotic combination the survival of small populations of antibiotic tolerant variants using an aminoglycoside alone may be a concern [31]. Recently it has been found that small populations of PA overexpressing tRNA pseudouridine synthase are tolerant to both TOB and gentamicin in concentrations well above MIC [40], although at very high concentrations even they can be eradicated [31].

Dual-antibiotic therapy, in contrast to monotherapy, may be beneficial as it provides multiple modes of action against Gram-positive and Gram-negative bacteria, has shown high efficacy in zone of inhibition tests and is less likely to result in spontaneous resistance due to single point mutations [41,42]. We found the combination of VAN + TOB released from HP-ALCSB significantly reduced the concentration of EF and PA compared to controls and either antibiotic alone. This may be due to synergistic activity of VAN in combination with TOB or a combined effect of organisms competing with each other for resources while also being confronted with dual-antibiotic therapy [43]. While the PA results did not show a significant reduction in zone sizes when comparing the use of TOB monotherapy to VAN + TOB dual therapy (Table 1, Figure 8), the combination of VAN + TOB reduced the number of surviving tolerant variants by almost 50% (Table 1, Figure 9). Although our study relied on plate counting to assess the viability of each species, in future studies a metabolic activity such as the MTT stain, which measures respiration rates, combined with confocal microscopy may provide finer granularity of the local effects of dual antibiotics on species in the dual species biofilm. 

In summary, this in-vitro study demonstrates the potential efficacy of using VAN + TOB dual-antibiotic therapy which may be beneficial in multispecies infections containing Gram-positive and Gram-negative pathogens and reducing the number of small population tolerant variant colonies. However, future study in-vivo is needed to comprehensively evaluate the use of dual-antibiotic therapy for multispecies infections.

## 5. Conclusions


(1)We were able to establish a Gram negative and Gram positive dual species biofilm of *P. aeruginosa* and *E. faecalis* on 316L stainless steel surfaces in approximately the same numerical proportions.(2)A dual combination of vancomycin and tobramycin released from high purity absorbable bone filler beads significantly reduced more biofilm bacteria than either antibiotic used alone.(3)Local release of multiple antibiotics resulting, in high concentrations and multiple mechanisms of action, may be more effective in treating PJIs involving multi-species biofilms with different antibiotic susceptibilities may be treated more effectively than from a single antibiotic alone.


## Figures and Tables

**Figure 1 microorganisms-11-02296-f001:**
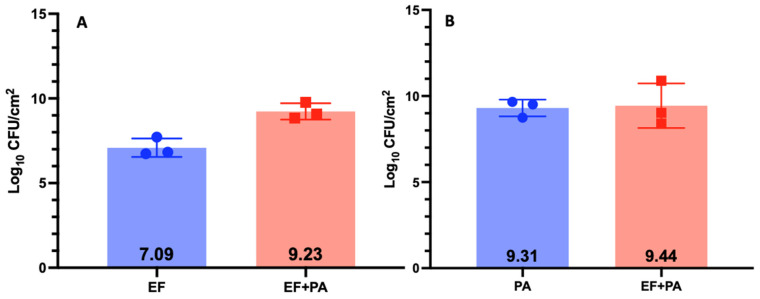
CFUs of EF-BS374 and PA-XEN41 3-day biofilm grown on 316L-SS coupons grown independently and together. Panel (**A**) shows EF grown as a monospecies and in an EF-BS374 + PA-XEN41 multispecies biofilm. Panel (**B**) shows PA grown as a monospecies and in an EF-BS374 + PA-XEN41 multispecies biofilm. N = 3, log_10_ of the geometric mean and 1 SD.

**Figure 2 microorganisms-11-02296-f002:**
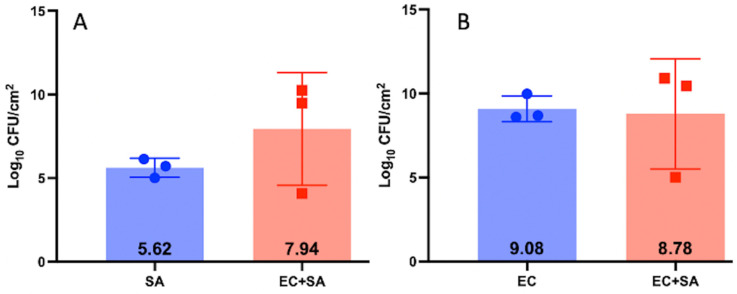
CFUs of SA-SAP231 and EC-BS420 3-day biofilm grown on 316L-SS coupons grown independently and together. Panel (**A**) shows SA grown as a monospecies and in a SA-SAP231 + EC-BS420 multispecies biofilm. Panel (**B**) shows EC grown as a monospecies and in a SA-SAP231 + EC-BS420 multispecies biofilm. N = 3, log_10_ of the geometric mean and 1 SD.

**Figure 3 microorganisms-11-02296-f003:**
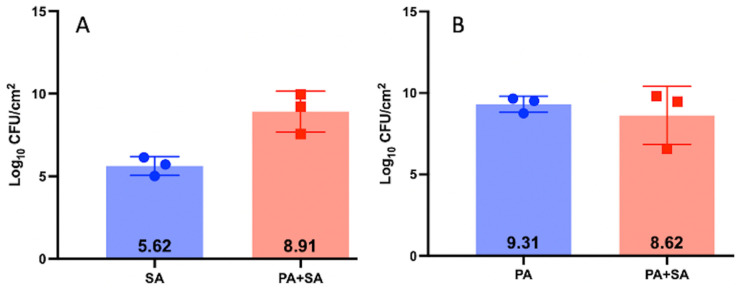
CFUs of PA-XEN41 and SA-SAP231 3-day biofilm grown on 316L-SS coupons grown independently and together. Panel (**A**) shows PA grown as a monospecies and in a PA-XEN41 + SA-SAP231 multispecies biofilm. Panel (**B**) shows SA grown as a monospecies and in a PA-XEN41 + SA-SAP231 multispecies biofilm. N = 3, log_10_ of the geometric mean and 1 SD.

**Figure 4 microorganisms-11-02296-f004:**
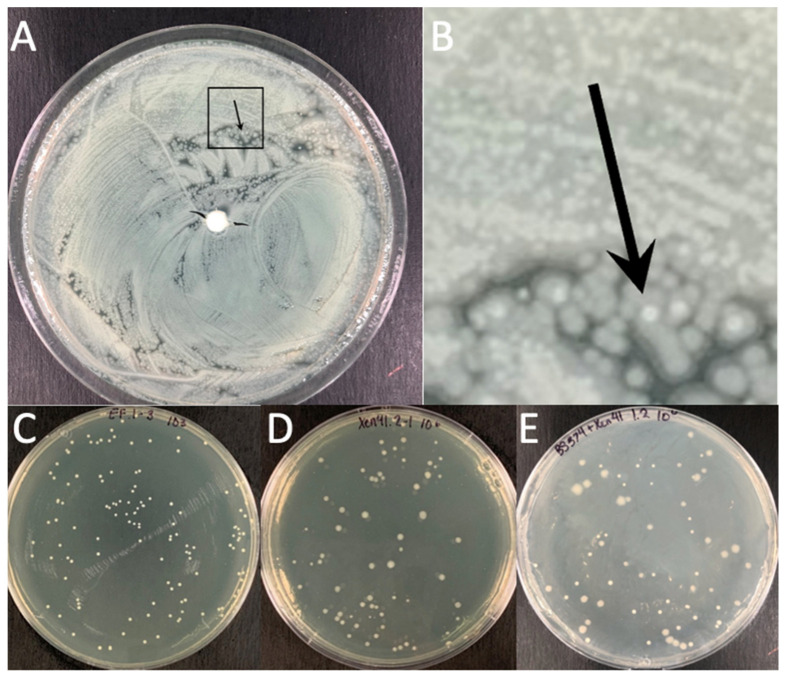
Representative plates of EF-BS374 and PA-XEN41 grown in a multispecies biofilm and a monospecies biofilm with an unloaded bead. PA-XEN41 forms a translucent green lawn whereas EF-BS374 form small circular opaque colonies growing within the lawn of PA-XEN41. The black arrow in Panel (**A**,**B**) points towards a single EF-BS374 colony. Panel (**A**) is the original image and Panel (**B**) is an enlarged portion of the original photo to better show the EF-BS374 and colony morphology. Panels (**C**–**E**) are all plates used to count the number of colonies within the multispecies biofilm and within the monospecies biofilms. Panel (**C**) is a plate of EF only. Panel (**D**) is a plate of PA only. Panel (**E**) is a plate with the multispecies biofilm containing EF + PA where PA colonies were confirmed as separate from EF colonies through bioluminescent imaging.

**Figure 5 microorganisms-11-02296-f005:**
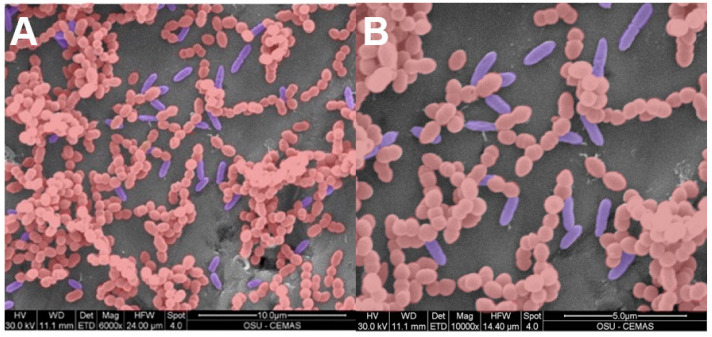
False coloured SEM images of a PA (blue rods) + EF (red cocci) multispecies biofilm grown on a 316L SS coupon. Panel (**A**) and Panel (**B**) represent 6000× magnification and 10,000× magnification respectfully. By SEM, EF appeared to dominate the biofilm by a factor of approximately 5. The presence of dividing rods and cocci was evidence that both species were actively growing together in the biofilm.

**Figure 6 microorganisms-11-02296-f006:**
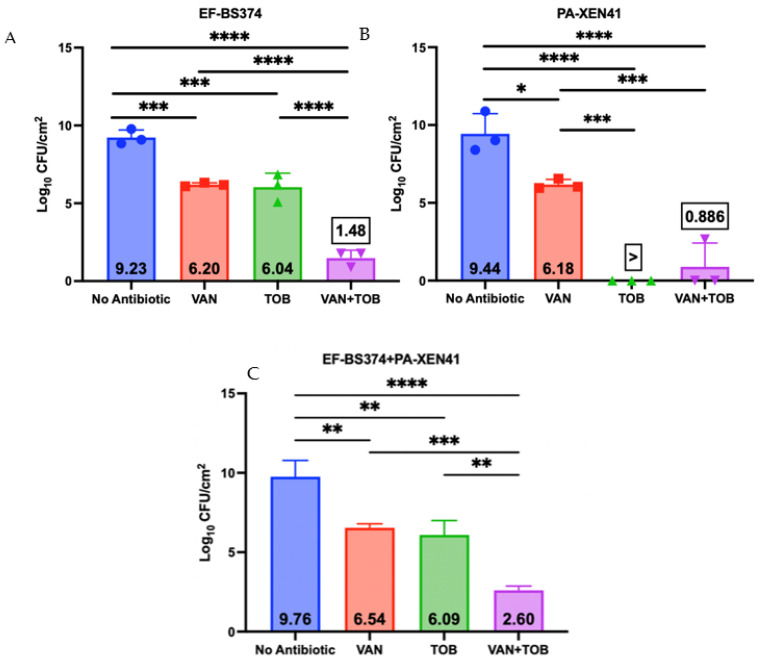
Killing efficacy of dual-species biofilms containing EF-BS374 and PA-XEN41 grown on 316L-SS coupons comparing no antibiotics, VAN alone, TOB alone or use of both VAN + TOB. Panel (**A**) demonstrates the EF portion of the multispecies biofilm, Panel (**B**) demonstrates the PA portion of the multispecies biofilm, and Panel (**C**) demonstrates the total multispecies biofilm. N = 3, log_10_ of the geometric mean ± SD. (* *p* < 0.05, ** *p* < 0.01, *** *p* < 0.001, **** *p* < 0.0001).

**Figure 7 microorganisms-11-02296-f007:**
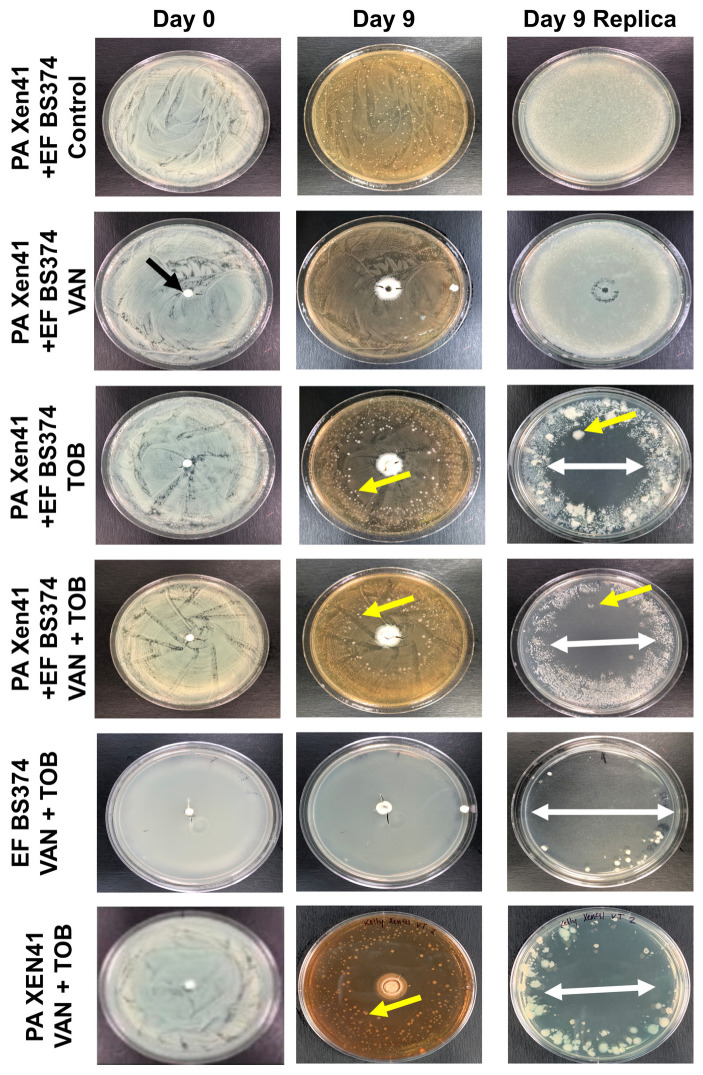
EF-BS374 and PA-XEN41 Light Images. Row 1 represents a control EF-BS374 and PA-XEN41 multispecies biofilm with no bead. Row 2 represents an EF-BS374 and PA-XEN41 multispecies biofilm treated with a HP-ALCSB loaded with 1000 mg VAN alone (black arrow) showing no zone of killing (ZOK) of PA but EF was eradicated. Row 3 represents an EF-BS374 and PA-XEN41 multispecies biofilm treated with a HP-ALCSB loaded with 240 mg TOB alone, showing a ZOK of both species (indicated by white arrow). An antibiotic tolerant variant of PA within the ZOK is indicated by the yellow arrow. The remaining background lawn at the periphery was dominated by EF (smaller colonies). Row 4 represents an EF-BS374 and PA-XEN41 multispecies biofilm treated with a HP-ALCSB loaded with 1000 mg VAN and 240 mg TOB. PA was eradicated but EF (small colonies) were still present in the remaining lawn at the periphery. Row 5 represents a single species EF-BS374 biofilm treated with 1000 mg VAN and 240 mg TOB. Row 6 represents a single species PA-XEN41 biofilm treated with 1000 mg VAN and 240 mg TOB. CSB were added at Day 0 and followed for 9 days before replica plating to observe viability on day 10. Representative antibiotic tolerant colonies growing in the ZOK are indicated by yellow arrows.

**Figure 8 microorganisms-11-02296-f008:**
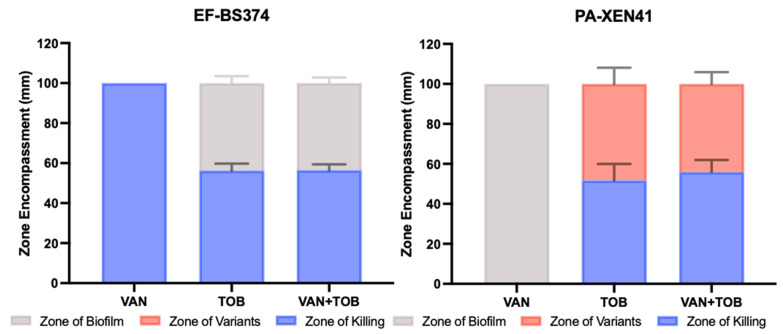
Zones of EF-BS374 and PA-XEN41 multispecies biofilms treated with a VAN, TOB or VAN + TOB HP-ALCSB. Zone of biofilm is the diameter of the unaffected lawn biofilm (grey), the zone of killing is the diameter where there was no growth at all (blue), and the zone of antibiotic tolerant variants was the diameter of the zone between the unaffected biofilm and complete killing (red). N = 3, Mean and 1 SD.

**Figure 9 microorganisms-11-02296-f009:**
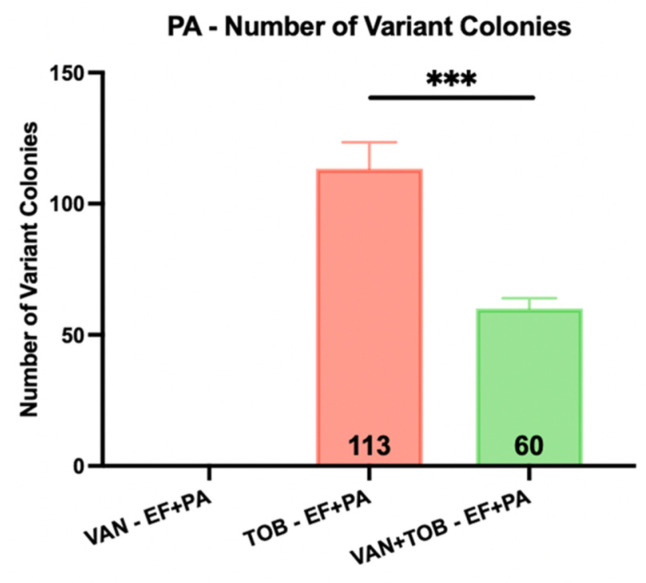
Number of PA antibiotic tolerant variant colonies emerging in the zone of inhibition of the EF + PA multispecies biofilms when treated by VAN, TOB or VAN + TOB. Individual colonies were observed and counted on day 10 of treatment after replica plating for viability. Mean and 1 standard deviation. (*** *p* < 0.001).

**Table 1 microorganisms-11-02296-t001:** Quantitative data used to generate Figure 7. This table also shows the number of variant colonies in addition to the size of the zone of variants. N = 3, Mean and 1 SD.

	VAN (EF-BS374)	VAN (PA-XEN41)	TOB (EF-BS374)	TOB (PA-XEN41)	VAN + TOB(EF-BS374)	VAN + TOB (PA-XEN41)	VAN + TOB–EF-BS374 Alone	VAN + TOB–PA-XEN41 Alone
Zone of Killing (mm)	100 ± 0	0 ± 0	56.3 ± 3.44	51.8 ± 8.10	56.6 ± 2.83	56.1 ± 5.88	72.8 ± 3.71	50.2 ± 1.84
Zone of Variants (mm)	0 ± 0	0 ± 0	0 ± 0	48.2 ± 8.1	0 ± 0	43.9 ± 5.88	0 ± 0	49.8 ± 1.84
Zone of Biofilm (mm)	0 ± 0	100 ± 0	43.7 ± 3.44	0	43.5 ± 2.83	0	27.2 ± 3.71	0 ± 0
# of Variants (count)	0 ± 0	0 ± 0	0 ± 0	113 ± 10.07	0 ± 0	60 ± 4	0 ± 0	175 ± 9.64

## Data Availability

Data supporting the reported results will be made available on request to P. Stoodley, paul.stoodley@osumc.edu.

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
