# Peer review of "Killing of a Multispecies Biofilm Using Gram-Negative and Gram-Positive Targeted Antibiotic Released from High Purity Calcium Sulfate Beads"

_microorganisms, 2023, doi:10.3390/microorganisms11092296_

Round 1

Reviewer 1 Report

In this manuscript, the action of vancomycin and tobramycin released from high purity calcium sulfate beads on biofilms of Gram positive and Gram negative bacteria was studied.

The following comments are made:

1. Line 70. What are the characteristics of clinical isolates? Put this

2. Line 85. In the combination of cells for biofilm, how many cells of each species do you put? put the amount

3. Line 98. How can you identify cells by: which were identified based on size, shape, color?

4. Line 144. Put what concentrations.

5. Figures 1, 2 and 3. Why did you use log and not cfu/mL?

6. Line 204. The proportion of species in the biofilm seems not to be the same. How do you explain this? Did you put the same amount of CFU for each species? Why is there more of one than the other? Put it up for discussion

7. Line 230. Again section 3.2? Correct

8. Figure 7. Can't see clearly getting bigger.

9. Table 2 is missing.

10. Figure 9 is not clear.

11. Lines 243-246. To explain better.

12. Why didn't you test a biofilm system with the three types of bacteria?

13. Why did you no longer use Staphylococcus aureus in the challenge experiments?

14. Why didn't you use plate biofilm formation to see the activity?

15. Missing conclusion section. Put it.

Minor editing of English language

Author Response

Reviewer 1

  1. Line 70. What are the characteristics of clinical isolates? Put this

Response: These strains have not been fully characterized and we apologize that the reference list where we reference the clinical case that they were isolated from and a subsequent study showing that they could form a multi species biofilm have been referenced (Refs 18 and 19).

  1. Line 85. In the combination of cells for biofilm, how many cells of each species do you put? put the amount

Response: ~2.5x105 , this has been added on line 86.

  1. Line 98. How can you identify cells by: which were identified based on size, shape, color?

Response. We selected the bacteria pairs based on the fact that they have different morphology differences (rods vs cocci) so that they could be readily distinguished in SEMs and confocal without the need for specific staining. This has now been added (Lines 73 to 74). They could be distinguished on an agar plate by bioluminescence as stated on line 65. The color and shape are mentioned in the results, lines 220 – 221.

  1. Line 144. Put what concentrations.

Response: Apologies, this was CFU/cm2, we have now added “/cm2”. This is correct in the graph axis.

  1. Figures 1, 2 and 3. Why did you use log and not cfu/mL?

Because CFU data often covers orders of magnitude it often does not conform to a normal distribution. It is common to perform a Log10 transformation to provide a more accurate representation of the central tendency – this is the Log10 of the geometric mean. We have added this rationale to the statistics section (Line 154. We have also corrected our text from “geometric mean”, to “log10 of the geometric mean and 1 SD”.

  1. Line 204. The proportion of species in the biofilm seems not to be the same. How do you explain this? Did you put the same amount of CFU for each species? Why is there more of one than the other? Put it up for discussion

Response: We did use the same starting concentrations (see response to comment 2). Presumably differences in the biofilm were due to different levels of fitness by the two species which were competing for the same space and nutrient source. We have added this text to the discussion” Although the numbers of EF and PA in the biofilm were on the same order of magnitude (9.23 and 9.44 log10 CFU/cm2 respectively, PA was in greater abundance than EF, 2.7x10^9 compared to 1.7 x10^9. This difference may be explained by different fitness levels in competition for space and nutrients”. Lines 266-268.

  1. Line 230. Again section 3.2? Correct

Response: This has been corrected.

  1. Figure 7. Can't see clearly getting bigger.

Response: Apologies but we have gone through the caption and all the references to Fig 7 and can not see where we make the statement of something getting clearly bigger. We presume the reviewer is referring to a zone of inhibition or colony size but cant be sure.

  1. Table 2 is missing.

Apologies, there is no Table 2, this should have been called out to Table 1, we have corrected.

  1. Figure 9 is not clear.

Response: We are not sure what is not clear. The bar chart itself appears to be fine so we assume it is the caption that is confusing. This is the number of PA antibiotic tolerant colonies that appear in the zone of inhibition. We have edited the text to hopefully make it clearer. Also in the main text we refer to the method we use in a previous paper (Sindeldecker et al.).  

  1. Lines 243-246. To explain better.

Response: It appears that the line numbers the reviewer is referring to are out of synch with the submitted word doc for some reason. If the reviewer is referring to the caption to Fig 9 please see our previous response to comment 10.

  1. Why didn't you test a biofilm system with the three types of bacteria?

Response: We used a combination of a Gram positive cocci and a Gram negative rod in a dual species in order to demonstrate the effect of the use of multiple antibiotics on strains with different antibiotic susceptibilities and as a simple model to reflect a multi species biofilm when both Gram positive and negative bacteria are found in an orthopedic infection. In addition to the increased complexity of adding a third member of the community we would have difficulty distinguishing them on the basis of bioluminescence and cell morphology which would have added further complexity. Since dual species composed of Gram positive and Gram negative species are commonly used in biofilm studies, for the sake of brevity we have not added the rationale for not using a 3 species biofilm to the methods or discussion.

  1. Why did you no longer use Staphylococcus aureusin the challenge experiments?

Response: The rationale for selecting our dual species was provided in section 2.2 and in the results section 3.1, which was based on our preliminary experiments showing that PA and EF populated the biofilm in relatively equal numbers.

  1. Why didn't you use plate biofilm formation to see the activity?

Response: We used stainless steel as it is commonly used for orthopedic devices (see line 74). We did use the dual species plate biofilm which we challenged by placing a bead as shown in Fig 7.

  1. Missing conclusion section. Put it.

Response: We have now added conclusions:

  1. Conclusions

1) A Gram negative and positive dual species biofilms of P. aeruginosa and E. faecalis established biofilms on 316L stainless steel surfaces in approximately the same numerical proportions.

2) SEM images suggested that the species were interspersed together within the biofilm rather than forming separate aggregates of each species.

3) A dual combination of vancomycin and tobramycin released from high purity absorbable bone filler beads significantly reduced more biofilm bacteria than either antibiotic used alone.

4)   A dual combination of vancomycin and tobramycin released from high purity absorbable bone filler beads significantly reduced the sub population of P. aeruginosa antibiotic tolerant variants than tobramycin alone.

5) Even though P. aeruginosa is resistant to vancomycin in the concentration range used in routine clinical microbiology susceptibility assays we found a significant reduction with vancomycin alone, possibly explained by the very high local concentrations achievable by local release from the beads, not attainable from IV administration, and microbial stress from nutrient starvation within the biofilm.

  1. Local release of multiple antibiotics resulting, in high concentrations and multiple mechanisms of action, may be more effective in treating PJIs involving multi-species biofilms with different antibiotic susceptibilities may be treated more effectively than from a single antibiotic alone.

Reviewer 2 Report

The manuscript entitled  «Killing of a Multispecies Biofilm Using a Gram-Negative and  Gram-Positive Targeted Antibiotic Released from High Purity  Calcium Sulfate Beads  is devoted to in-vitro study of a multispecies biofilm growth with antibiotic-loaded calcium sulfate beads (HP-ALCSB) containing vancomycin (VAN) and tobramycin (TOB) in combination and alone». The manuscript is devoted to an important and relevant topic Periprosthetic joint infections (orthopedic infections) and in general it seemed interesting and relevant to me, corresponding to the aims and scopes of the Microorganisms journal. It seems to me that there is a little fundamental scientific novelty in the work, but there is an important applied significance.

General remarks

1.Since the authors propose a new solution to the use of antibiotics, the introduction should describe in more detail the novelty of the study and dwell on works that have adopted similar approaches to the use of antibiotics.

2.to assess fouling on the surface of materials, I would advise the authors to use the MTT test, which can quite simply give information about the respiratory activity of microorganisms.

3.Micrographs obtained by Scanning electron microscopy (SEM) show microorganisms in two colors. The authors should explain in more detail how the color photographs were obtained.

4.I have no complaints about the text itself, it is written quite clearly, well illustrated.

However, the text lacks a conclusion. it should clearly show what is done for the first time and what are the most important results obtained. Without them, the text resembles a technical report.

Author Response

Rev 2

The manuscript entitled  «Killing of a Multispecies Biofilm Using a Gram-Negative and  Gram-Positive Targeted Antibiotic Released from High Purity  Calcium Sulfate Beads  is devoted to in-vitro study of a multispecies biofilm growth with antibiotic-loaded calcium sulfate beads (HP-ALCSB) containing vancomycin (VAN) and tobramycin (TOB) in combination and alone». The manuscript is devoted to an important and relevant topic Periprosthetic joint infections (orthopedic infections) and in general it seemed interesting and relevant to me, corresponding to the aims and scopes of the Microorganisms journal. It seems to me that there is a little fundamental scientific novelty in the work, but there is an important applied significance.

Response: We thank the reviewer for seeing translational relevance to our lab study.

General remarks

1.Since the authors propose a new solution to the use of antibiotics, the introduction should describe in more detail the novelty of the study and dwell on works that have adopted similar approaches to the use of antibiotics.

Response: We have now added our rationale and novelty for the experiment in the introduction, lines 55-60.

2.to assess fouling on the surface of materials, I would advise the authors to use the MTT test, which can quite simply give information about the respiratory activity of microorganisms.

Response: We agree that the MTT test is useful for assessing metabolic activity and compliments CFU data which just provides viability status. Thus bacteria that might not be metabolically active in situ might still be viable (i.e. dormant or persister cells). MTT also has the advantage with confocal microscopy where it can give spatial distribution within the biofilm. We will consider MTT in future studies (and indeed have used it before) and have written a brief mention of how our study could be improved in the discussion (Lines 297-299).

3.Micrographs obtained by Scanning electron microscopy (SEM) show microorganisms in two colors. The authors should explain in more detail how the color photographs were obtained.

Response: Bacteria in the SEM micrographs were identified by eye based on size (rods and cocci) and shape (~ 1 µm dia. and (~ 1 µm width and 2-3 µm length for the rods) and false colored using photo-editing software (Adobe Photoshop version 22.4.2) to allow better contrast with the background. This has been added to the methods, lines 105-107.

4.I have no complaints about the text itself, it is written quite clearly, well illustrated.

However, the text lacks a conclusion. it should clearly show what is done for the first time and what are the most important results obtained. Without them, the text resembles a technical report.

Response: We have now added a conclusion section as both reviewers had requested.

Round 2

Reviewer 1 Report

1. Explain in the text why S. aureus was no longer used

2. The photographs in figure 7 are very small and it is not possible to clearly differentiate the formation or not of biofilm, so it is suggested to increase the size of the photos.

3. The conclusions section is a section where the conclusion of your work is finalized, not the conclusions of each of the experiments. Those can be used for discussion. Therefore, the conclusion regarding the aim of the work should be put in the conclusions section.

The quality of English is acceptable

Author Response

    1. Explain in the text why S. aureus was no longer used

    Response: We understand that S. aureus is an important pathogen in PJI and we were hoping that we could grow a dual species biofilm which included S. aureus. However, as we had previously explained in our previous response, and made edits in the text to make it clearer, in our rationale for our final choice, we had a set of criteria for selection. 1) to include a Gram negative and positive species to provide different antibiotic susceptibilities, 2) a rod and a cocci so that we could differentiate them by SEM, 3) each species to establish in the biofilm in approximately equal numbers. We have now stated thess in the revision.  

    1. The photographs in figure 7 are very small and it is not possible to clearly differentiate the formation or not of biofilm, so it is suggested to increase the size of the photos.

    Response: We have now increased the size and resolution of Fig. 7. We have also removed the third column (Day 5) since we thought it added complexity without adding to interpretation. In addition we have added arrows to indicate features discussed in the caption and have edited the caption to make it more understandable to the reader. We also added a supplemental figure showing how we used bioluminescence to distinguish P. aeruginosa and added the line “as well as being bioluminescent (Supplemental figure S1) to the main text, line 224.

    1. The conclusions section is a section where the conclusion of your work is finalized, not the conclusions of each of the experiments. Those can be used for discussion. Therefore, the conclusion regarding the aim of the work should be put in the conclusions section.

    Response: We have now removed two conclusions and brought up in the discussion. The remaining 3 conclusions are aligned with the aims of the experiments.

Reviewer 2 Report

having carefully read the revised manuscript, I can say that my remarks have been taken into account. I believe that the text can be published in this form.

Author Response

We thank the reviewer for their time and useful comments from the initial review.